# Assessment of Direct Normal Irradiance Forecasts Based on IFS/ECMWF Data and Observations in the South of Portugal

**João Perdigão** [1,*] **, Paulo Canhoto** [1,2] **, Rui Salgado** [1,2] **and Maria João Costa** [1,2]

1   Instituto de Ciências da Terra, Universidade de Évora, Rua Romão Ramalho 59, 7000-671 Évora, Portugal; canhoto@uevora.pt (P.C.); rsal@uevora.pt (R.S.); mjcosta@uevora.pt (M.J.C.)
2   Departamento de Física, Escola de Ciências e Tecnologia, Universidade de Évora, Rua Romão Ramalho 59, 7000-671 Évora, Portugal
*   Correspondence: perdi.j@gmail.com

**Abstract:** Direct Normal Irradiance (DNI) predictions obtained from the Integrated Forecasting System of the European Centre for Medium-Range Weather Forecast (IFS/ECMWF) were compared against ground-based observational data for one location at the south of Portugal (Évora). Hourly and daily DNI values were analyzed for different temporal forecast horizons (1 to 3 days ahead) and results show that the IFS/ECMWF slightly overestimates DNI for the period of analysis (1 August 2018 until 31 July 2019) with a fairly good agreement between model and observations. Hourly basis evaluation shows relatively high errors, independently of the forecast day. Root mean square error increases as the forecast time increases with a relative error of ~45% between the first and the last forecast. Similar patterns are observed in the daily analysis with comparable magnitude errors. The correlation coefficients between forecast and observed data are above 0.7 for both hourly and daily data. A methodology based on a new DNI attenuation Index (DAI) was developed to estimate cloud fraction from hourly values integrated over a day and, with that, to correlate the accuracy of the forecast with sky conditions. This correlation with DAI reveals that in IFS/ECMWF model, the atmosphere as being more transparent than reality since cloud cover is underestimated in the majority of the months of the year, taking the ground-based measurements as a reference. The use of the DAI estimator confirms that the errors in IFS/ECMWF are larger under cloudy skies than under clear sky. The development and application of a post-processing methodology improves the DNI predictions from the IFS/ECMWF outputs, with a decrease of error of the order of ~30%, when compared with raw data.

**Keywords:** Direct Normal Irradiance (DNI); IFS/ECMWF; forecast; evaluation; DNI attenuation Index (DAI); bias correction

## 1. Introduction

Solar energy is becoming a crucial renewable resource in modern societies, contributing to the sustainability of the planet with the mitigation of greenhouse gas emissions by reducing the consumption of coal or fuel oil for electricity production; however, the availability of solar resources over time at a given region of interest determines the cost/benefit of solar power plants implementation. Since the temporal series of solar radiation measurements are spatially limited, and thus scarce and sometimes inexistent, the prediction and validation of solar resource is a key factor for such enterprises.

Several researchers have estimated the potential of renewable energies like wind or solar radiation for electricity or thermal energy production around the world; for example, in Europe and Africa [1],

in Chile [2], in Iberian Peninsula [3], in United Kingdom [4], and Spain [5]. Solar power is a very promising energy source in the Iberian Peninsula (IP) and strong growth is expected in this area. In the IP there are multiple options for using renewable energy (solar, wind, and hydro) to generate electricity; however, the solar resource is high throughout the year [6–8].

Concerning solar energy, there are two main ways of converting solar energy into electricity: Photovoltaic (PV) and Concentrating Solar Power (CSP). The PV panels convert either direct and diffuse solar irradiance, while the CSP technology only concentrates the Direct Normal Irradiance (DNI). The focus of this work is on the prediction of DNI, because of its use in CSP plant management. The forecast of global solar radiation (direct + diffuse) for the same region was addressed, for example, in [6] and [9].

There are several approaches to predict solar irradiance such as Numerical Weather Prediction (NWP), Cloud Motion Vector (CMV), statistical time series analysis, and other methods [7,10,11]. In the last years, one of the major research challenges for the use of NWP in solar energy applications is the DNI forecast, aiming at the development and increase of CSP installed capacity and operation management. CSP requires the knowledge of DNI for specific sites [12,13], and one of the difficulties is the need to forecast the DNI with several days ahead to increase efficiency and minimize the operational costs of the power plants [14,15]. For instance, Casado-Rubio et al. [16] proposed a simple methodology to obtained DNI forecast, based on Weather Research and Forecasting model (WRF model) and a radiative transfer simulation (for 1-day forecast) and found that this procedure can be used as a diagnostic tool for operational power plants.

Until recently, DNI measurements were not available in many places or with long series, and this variable was not a direct output of NWP models. Currently, the Integrated Forecast System of the European Centre for Medium-Range Weather Forecasts (IFS/ECMWF) provides the direct normal irradiance as an output; however, the use of NWP models in the DNI forecast is still not perfect and requires Multiple Output Statistic (MOS) methodologies [13]. Lopes et al. [17] used the IFS global model of ECMWF to assess DNI for short-term (24 h) in the south of Portugal and found relative differences in the range ~7% to ~12% on an annual basis between predictions and observations at ground-based stations. Lara-Fanego et al. [18] found a relative root mean square error of 60% for hourly DNI forecasts in Spain for all sky conditions, using the Advanced Research Weather Research and Forecasting model (WRF). Troccoli and Morcrette [19] analyzed the direct solar radiation data using two different radiation schemes of the IFS/ECMWF for four ground-based measuring stations in Australia and found mean absolute errors between 18% and 45% and correlation coefficients between 0.25 and 0.85. In that work, the usage of a post-processing bias correction improved results, resulting in mean absolute errors between 10% and 15% and correlation coefficients of about 0.9. Ruiz-Arias et al. [7] also found better results for DNI forecasts from the WRF model by using a post-processing algorithm. Law et al. [13] present a comprehensive review of DNI forecast obtained from several methods and some examples of DNI forecast accuracies are presented. According to Vick et al. [20], most studies on DNI models have assessed the annual and hourly mean bias and root mean square errors between measured and DNI models; however, according to the same authors, the accuracy of monthly and daily direct normal irradiation forecasts should also be assessed to detect gaps in DNI modeling that may be improved and correlated with sky conditions, time of the year, or location.

Since there are several on-going projects in Portugal to explore the solar resource, it is imperative to carry out studies that help understand the errors associated with direct normal irradiance predictions over several days ahead.

The Portuguese Institute for Sea and Atmosphere (IPMA), the meteorological Portuguese authority, uses the ECMWF global model predictions as the main forecast tool. Comparisons between operational global NWP models show that ECMWF over Europe is the best [21]. Good numerical predictions of the near-surface weather conditions presuppose a good representation of the surface radiative balance; however, a correct forecast of the global irradiance does not necessarily mean an accurate partition between direct and diffuse components, as this partition is not essential to solving the surface balance.

In response to growing demand from the solar energy market, the ECMWF has recently (2015) started to include DNI among the available predicted variables. These forecasts are likely to be used by solar plants in southern Portugal.

The main objective of this study was to assess the performance of the IFS/ECMWF global model (CY45R1 cycle—released at 5 June 2018) to predict DNI in the south of Portugal, by comparing its results with observational data of Évora station, on an hourly and daily basis and for various forecasting horizons (up to four days ahead). We present a method to predict sky conditions based on observational DNI data. A post-processing methodology was also tested to minimize the bias in the IFS/ECMWF model.

The paper is organized as follows: Section two describes the data and methods used in this study, the performance assessment of the ECMWF forecasts is presented and discussed in Section 3, and finally, conclusions are provided in Section 4.

## 2. Materials and Methods

### 2.1. DNI Observational Data

The measurements used in this study were obtained from the observatory of Atmospheric Sciences located at the University of Évora (38.57° N, 7.9° W, 293 m a.m.s.l.). DNI was measured using a first-class pyrheliometer (Kipp & Zonen, model CHP01) [22], following the World Meteorological Organization (WMO) [23] and the International Organization for Standardization (ISO), the 9060:1990 standard [24]. This model of pyrheliometer was designed to measure the solar irradiance with an opening half-angle of 2.5°.

A period of one year of DNI measurements was used in this study, from 1 August 2018 until 31 July 2019. The sensor output was sampled every 5 s and one-minute mean, minimum, maximum, and standard deviation values were recorded. Hourly values were then computed by averaging one-minute values when the number of records for that hour corresponds to at least fifty minutes. The data for solar zenith angles above 89° (twilight and nighttime) were not considered and thus removed from the analysis. The daily mean was computed using a similar methodology of that used by Troccoli and Morcrette [19], i.e., if one or more hourly values are not present on a given day, then that day is not used in the analysis.

All instruments of this measuring station were subject to maintenance and cleaning procedures following the recommendations of the World Meteorological Organization and data was subject to BSRN (Baseline Surface Radiation Network) quality filters [25] based on physically possible and extremely rare values.

In this work, the seasons were defined according to the WMO nomenclature i.e., winter (December–January–February: DJF), spring (March–April–May: MAM), summer (June–July–August: JJA), and autumn (September–October–November: SON).

### 2.2. DNI Forecast Data

Predicted DNI from IFS/ECMWF, from 1 August 2018 until 31 July 2019, was obtained with a resolution of 0.125 × 0.125 (lat × lon grid). The forecast data were provided with an hourly time step for the first three days and with a three-hour time step for the 4th day. In this work, forecasts were separated into four intervals: day_0 (1st day); day_1 (2nd day); day_2 (3rd day); day_3 (4th day). The predicted accumulated solar irradiation in hour time steps for the entire forecast horizon was converted into hourly mean irradiance values of DNI.

The shortwave radiation scheme of the IFS/ECMWF used in this study was the new radiation scheme implemented on 11 July 2017, called ecRad [26]. This scheme is faster than the previously McRad scheme [27] and can be executed more times during the forecast. This scheme computes the profiles of shortwave and longwave irradiances at half levels, and these are interpolated horizontally back onto the model grid using cubic interpolation [26]. The aerosol distribution was adapted from

Tegen et al. [28], using a climatology of six hydrophobic aerosol species as well as the newer climatology obtained from a reanalysis of the atmospheric composition produced by the Copernicus Atmosphere Monitoring Services (CAMS), with 11 hydrophilic and hydrophobic species [29].

According to Hogan and Bozzo [26], ecRad incorporates a method to represent longwave scattering of clouds, which leads to an improvement in forecast skills. The default ice optical properties were computed using the Fu scheme [30], but two additional schemes were available.

In the work by Hogan and Bozzo [26], it was possible to find the evolution of the ECMWF Radiation Scheme after 2000 and the options available. More details on the physical processes (and the options available) were reported in the IFS documentation on the ECMWF web page [31].

The nearest neighbor technique was used to select the forecast data for comparison with measurements. To assess forecasting accuracy, the observational data was compared with the forecasts for the nearest model grid point. Wild and Schmucki [32], made several statistical tests surrounding a grid point to analyze trends and the results showed that different grid points surrounding a given grid point (selected by a Lat/Lon value) do not differ significantly from each other in the majority of the cases.

### 2.3. Statistical Indicators for Model Ssessment

The quality of the DNI forecasts was evaluated against observational data using common statistical parameters as the Mean Bias Error (MBE), Mean Absolute Error (MAE), Root Mean Square Error (RMSE), and Correlation Coefficient (r). In this work, errors were calculated based on hourly, daily, and monthly mean values. Similar to the analysis presented by Nonnenmacher et al. [14] and Perez et al. [33], night-time values (zero solar irradiance) were excluded from the model assessment. The ratio (RSR) between the root mean square error and the observations standard deviation ($\sigma_{obs}$) was also determined.

These statistical parameters are defined as follows:

$$MBE = \frac{1}{N} \sum_{i=1}^{N} (m_i - o_i) \tag{1}$$

$$MAE = \frac{1}{N} \sum_{i=1}^{N} |m_i - o_i| \tag{2}$$

$$RMSE = \left[ \frac{1}{N} \sum_{i=1}^{N} (m_i - o_i)^2 \right]^{\frac{1}{2}} \tag{3}$$

$$r = \frac{\sum_{i=1}^{N} (o_i - \bar{o})(m_i - \bar{m})}{\left[ \sum_{i=1}^{N} (o_i - \bar{o})^2 \sum_{i=1}^{N} (m_i - \bar{m})^2 \right]^{\frac{1}{2}}} \tag{4}$$

$$RSR = \frac{RMSE}{\sigma_{obs}} = \frac{\left[ \frac{1}{N} \sum_{i=1}^{N} (m_i - o_i)^2 \right]^{\frac{1}{2}}}{\left[ \sum_{i=1}^{n} (o_i - \bar{o})^2 \right]^{\frac{1}{2}}} \tag{5}$$

where $N$ is the number of data points and $m$ and $o$ are the forecast and observed values, respectively. The MBE represents a systematic error between predicted and observational values, and the RMSE quantifies the spread in the distribution of errors. The MBE provides information on the underestimation (negative values) or overestimation (positive values) of forecasts using the measured values as reference. On the other hand, the RMSE is very sensitive to high magnitudes errors due to the higher statistical weight of large errors. The MAE represents the average magnitude of errors in a set of forecasts without considering their direction (bias) and gives the same weight to all errors (see as an example, Chai and Draxler [34]), i.e., it is less sensitive to large deviations. RMSE is one of the most relevant statistical parameters for solar power plant analysis (e.g., Landelius et al. [35]). For all statistical

parameters, the best results are obtained when values are equal or near zero, except for the correlation coefficient when values closer to one correspond to better performances. According to Moriasi et al. [36], RSR incorporates the benefits of error index statistics and includes a scaling/normalization factor, so that the resulting statistic and reported values can apply to various constituents. The RSR parameter varies between zero and a positive value with the values close to zero representing a better forecast simulation [36]. The same thresholds performance ratings as shown in Table 4 of the article of Moriasi et al. [36] are used here, i.e., values of RSR < 0.5 indicate the optimal performance rating while RSR > 0.7 represents unsatisfactory model performance rating.

### 2.4. Cloud Area Fraction and DNI Attenuation Index (DAI)

In most of the solar radiation studies, an index known as clear sky index or the clearness index is used to quantify the bulk atmosphere transmittance (see Iqbal [37], Lopes et al. [17], among others). This clearness index is defined as the ratio of global horizontal irradiance to extraterrestrial horizontal irradiance.

In this section, a new methodology is proposed to estimate the clearness of the atmosphere (termed DNI attenuation index-DAI) based exclusively on the observed direct normal irradiance. DNI varies during the day due to Sun–Earth geometry and atmospheric constituents, though the main factor of DNI variation is the cloud coverage, which can drastically reduce this component of solar radiation when the direct beam from the sun is intercepted, sometimes reaching a value of zero depending on the type of clouds. The DAI is an indicator of the cloud attenuation of DNI.

This method was based on the integration of the measured hourly mean values of DNI (see Figure 1a) obtained every day, for a given month, and it constitutes a measure of sky conditions for a particular month. This has the advantage of not relying on reanalysis, satellite data, or other products that could also be a source of bias.

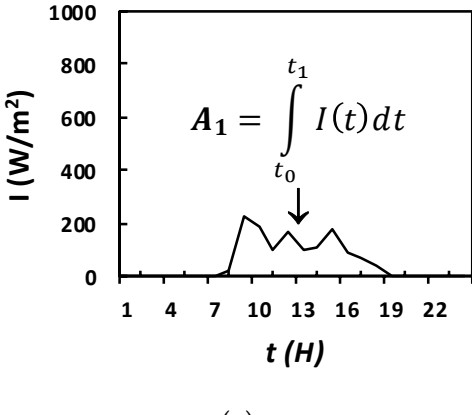 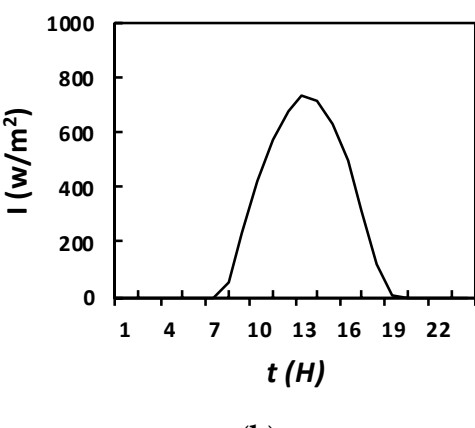

(**a**)                                                                                      (**b**)

**Figure 1.** Curves of hourly mean Direct Normal Irradiance (DNI) projected on the horizontal plane for two different days: (**a**) partially cloudy sky and (**b**) clear sky day. A is the area under the curve corresponding to the measured DNI (energy per unit area) and is obtained by numerical integration using the trapezoid rule.

In this way, a dimensionless quantity (in percentage) called DNI Attenuation Index (DAI) is defined as

$$DAI_i = \left(1 - \frac{A_i}{NF}\right) \times 100\% \tag{6}$$

where

$$A_i = \int_{t0}^{t1} I(t)dt \sim \frac{\Delta t}{2} \sum_{k=1}^{24} (I_{k-1} + I_k) \tag{7}$$

with $\Delta t = 3600$ s because a time step of one hour is used and NF is the normalization factor calculated as

$$NF = \max_{1 \le i \le n} (A_i) \tag{8}$$

in which $i$ is the number of the day of the month.

To obtain the DAI it is assumed that the maximum value of daily energy per unit area (integral) in a given month is interpreted as a clear sky day in that month and the DAI will take the value of zero for that particular day. Different normalization factors will be expected for different months, with higher values during the summer. Although DAI does not allow us to effectively distinguish the contribution of aerosols or cloud cover to DNI variations, it provides a clear idea of the transparence of the atmosphere for a specific day and it hints at the identification of a clear day (or clearness of atmosphere) from an overcast day or extreme aerosol event. The DAI varies between zero (clear sky day) and one (overcast sky).

The relation between DAI and cloud fraction (in oktas) was established through three classes of days [23]: class I – clear sky day (0–2 oktas; DAI< 31.25%); class II – partially cloudy skies (3–5 oktas; 31.25% ≤DAI< 68.75%) and class III as cloudy skies (6–8 oktas; 68.75% ≤ DAI ≤ 100%), in the same way as presented in Table 1 of the article of Jafariserajehlou et al. [38].

**Table 1.** Statistical indicators of comparison between observed and predicted hourly mean Direct Normal Irradiance (DNI) for the entire period (1 August 2018–31 July 2019). Bold values mean the best score.

| Day | MBE (W/m$^2$) | MAE (W/m$^2$) | RMSE (W/m$^2$) | $r$ |
|:---:|:---:|:---:|:---:|:---:|
| 0 | 13.54 | **136.80** | **195.41** | **0.84** |
| 1 | 15.03 | 146.35 | 210.60 | 0.81 |
| 2 | 17.273 | 154.97 | 224.02 | 0.78 |
| 3 | **1.048** | 197.88 | 267.25 | 0.70 |

The total cloud area fraction obtained from the Clouds and the Earth's Radiant Energy System (CERES) radiometer, combined with the Moderate Resolution Imaging Spectroradiometer (MODIS), both onboard the Terra and Aqua satellites, was also considered in this work for assessment of DAI estimates. The CERES–MODIS cloud mask data were obtained monthly (CERES_SYN1deg_Ed4.1) for the period available for this study (August 2017 until May 2019) and from CERES portal (see ceres.larc.nasa.gov). The cloud area fraction consists of the percentage of cloudy pixels identified in areas of 1° × 1° [39].

*2.5. Post Processing Correction*

A linear least square statistical method for bias correction to correct daily direct solar radiation values obtained from IFS/ECMWF was tested. This method is the simplest post-processing technique and has been applied in several studies over the past years (see for example Polo et al. [40]). Mejia et al. [41] found that MOS linear fit procedure outperformed the quantile–quantile mapping (Q–Q).

The linear regression parameters were computed for each month using forecast and observed daily values for the period of 1 August 2017 until 31 July 2018. The correction parameters are obtained, for each month, using the linear equation

$$y^i_{model,j} = m_j x^i_{obs,j} + b_j$$
$$i = 1, \ldots 28/30/31; \; j = 1, \ldots 12 \tag{9}$$

where $x_{obs}$, $m_j$ and $b_j$ are, respectively, the observed DNI values, the slope of the fitted line, and the intercept.

The regression parameters were used to correct the IFS/ECMWF forecasts for the following year—the period of analysis (01/08/2018 until 31/07/2019)—using the following equation [42],

$$y^i_{BC_{model},j} = y^i_{model,j} - \left[ (m_j - 1) x^i_{obs,j} + b_j \right]$$
$$i = 1, \ldots 28/30/31; j = 1, \ldots 12$$

(10)

## 3. Results and Discussion

### 3.1. Assessment of Hourly and Daily DNI Forecasts

As an example, Figure 2 shows the time series of predicted hourly mean DNI during four consecutive days and the corresponding observed values for two selected cases, one in JJA, other in SON: forecasts issued on 1 March 2017 00:00 and on 27 November 2017 00:00.

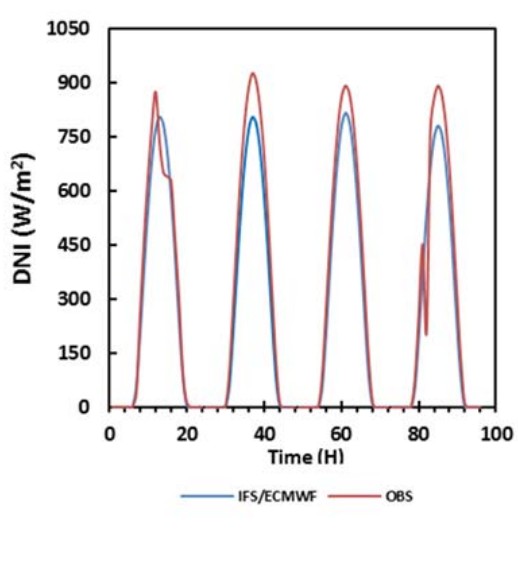
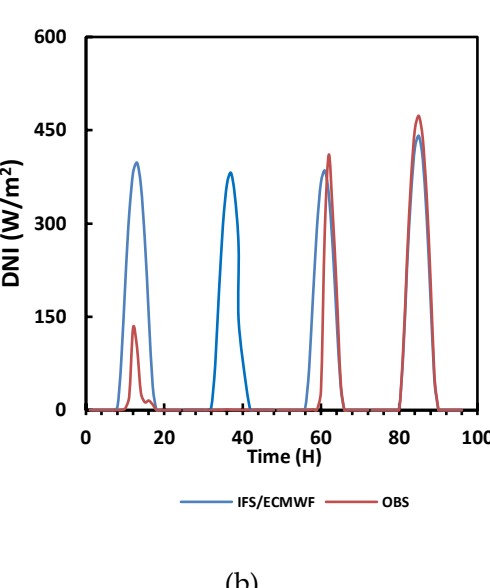

**Figure 2.** Example of four consecutive days of observed (red line) and forecasted (blue line) hourly mean DNI in Évora starting at (**a**) 1 August 2017 00:00 and (**b**) 27 November 2017 00:00.

The IFS/ECMWF forecasts have similar behavior to that of observational DNI for the two selected cases, with a fairly good agreement, especially in the case of August (Figure 2a). Figure 2b shows a partially cloudy day (day_0) and a cloudy day (day_1), making it evident that the model did not predict clouds correctly on November 28 since observational data clearly shows an overcast day. Another interesting feature in Figure 2 is that the IFS/ECMWF scheme slightly underestimated the DNI in the Summer case (Figure 2a) and overestimated it in the Autumn case (Figure 2b) in the case of a partly cloudy day. It is important to note that the example presented in Figure 2 is simply a selected example.

Figure 3 shows the comparison between ground-based measurements of hourly mean DNI and forecast data obtained from IFS/ECMWF for the entire period of study and the four forecasted days.

As expected, the errors associated with the hourly DNI forecast are quite significant with a strong scatter around y = x line (dashed line). The slope of the regression line indicates the quality of the forecasts and it is possible to conclude that the DNI IFS/ECMWF forecast is reasonable for the first three days ahead since the density of points is higher around the y = x line (dashed line in Figure 3). The worst forecast is for day_3. From Figure 3, it is also possible to verify that IFS overestimates DNI for lower values and underestimates DNI for higher irradiance values. This underestimation can be explained by the use of a constant monthly aerosol climatology in the IFS/ECMWF as argued

by Lopes et al. [17], concluding that the model tends to underestimate DNI under very clear sky atmospheric conditions, when the actual aerosol concentrations are below mean values.

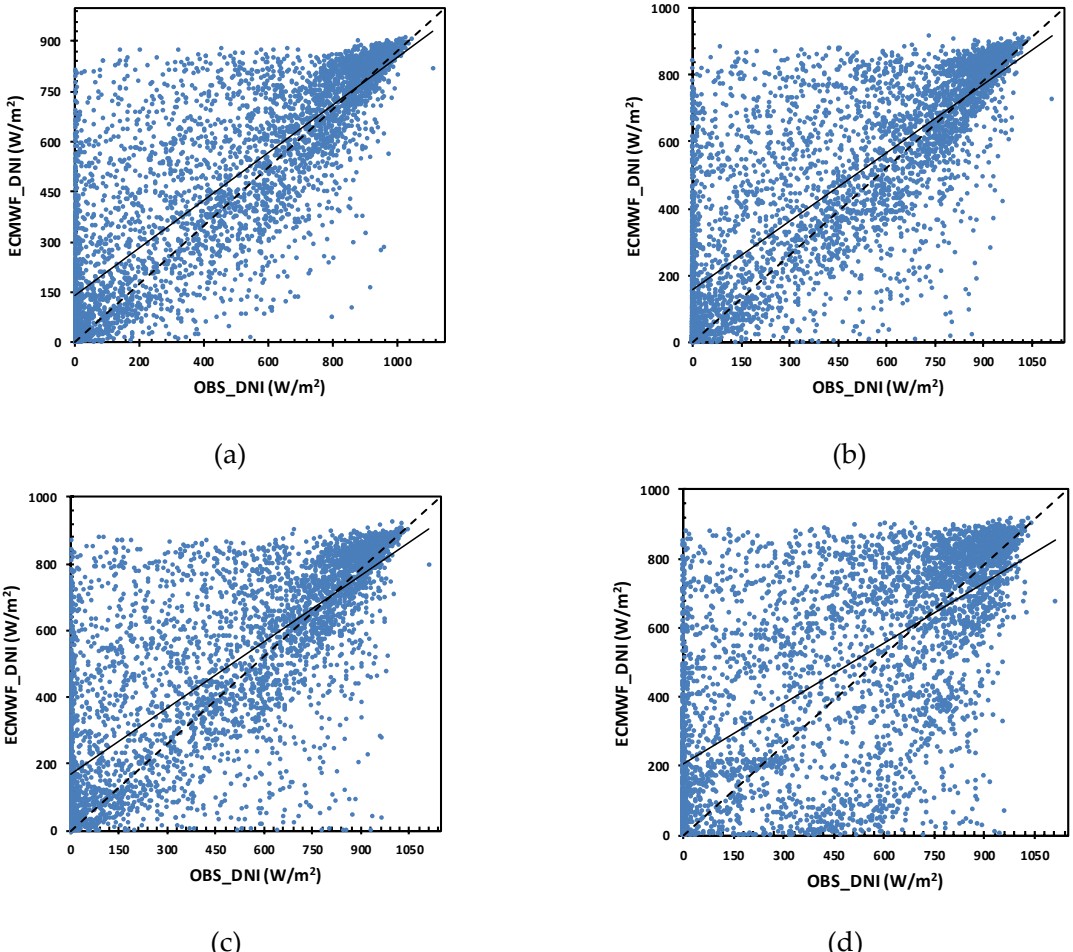

**Figure 3.** Scatter plots of predicted vs. measured hourly mean DNI for: (**a**) day_0, (**b**) day_1, (**c**) day_2, and (**d**) day_3, during the entire period considered. The dashed line represents the y = x line and the solid line is the least-squares regression fit.

The statistical errors, on an hourly basis, are presented in Table 1.

The assessment between datasets shows that errors increase from the first day of the forecast to the last day; day_0 exhibited the best performance with the lowest errors. MBE between calculated and measured DNI is smaller than 18 W/m$^2$. Regarding the MAE and RMSE, their values increase from day_0 to day_3 (fourth day of the forecast) with a difference between them of ~61 W/m$^2$ (~45%) and ~72 W/m$^2$ (~37%), respectively. High correlation coefficients (r ≥ 0.70) are obtained between the observations and forecasts for all forecast horizons (see Table 1).

The boxplots of Figure 4 show the MBE, MAE, RMSE, and correlation coefficient based on hourly values for each forecast day and the entire period of data (365 days).

MBE indicates a slight overestimation of hourly mean DNI for the majority of the forecast days (>50%) in the period. The length of the Interquartile Range (IQR) is a measure of the relative dispersion of a dataset and Figure 4a shows a similar length, in IQR, for the first two days of forecasts with values in (~−80; ~100 W/m$^2$). On the other hand, the difference between the IQR of the first forecast day and the last one (day_4) in the same plot is about 24%. Concerning MAE and RMSE, as expected, a similar pattern like MBE was found, with errors increasing as the lead time of the forecast increases, and a relative percentage error, relatively to the mean, between day_0 and day_3, for both parameters, of the order of 30%. It is worth noting that a significant number of outliers exist after the second day of

forecasting. As for the correlation coefficients, these values indicate a good forecast performance with the best results obtained for day_0 with the highest median value of ~0.98 (Figure 4d). The correlation coefficient (r) presents a good performance for all forecast days in the analysis.

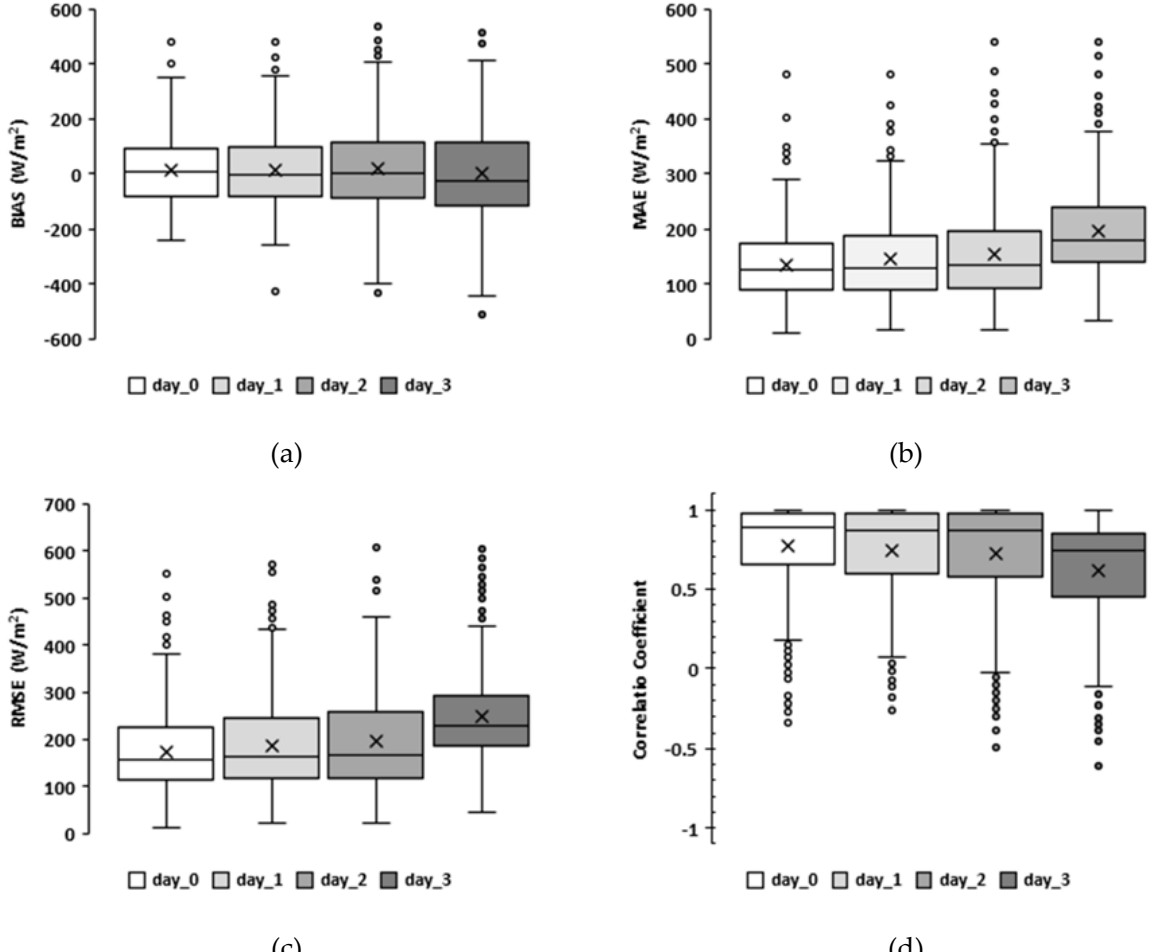

**Figure 4.** Boxplots of statistical indicators based on hourly values for (**a**) Mean Bias Error (MBE), (**b**) Mean Absolute Error (MAE), (**c**) Root Mean Square Error (RMSE), and (**d**) Correlation Coefficient (r), for different days ahead of forecast. The crosses represent the mean value of the sample, the horizontal solid line within the box represents the median, and the bottom and top of the boxes indicate the first and third quartiles, respectively. Boxes correspond to the Interquartile Range (IQR) where 50% of the data is located. The circles represent the outliers, and the lower and upper ends of the whiskers are the minimum and maximum values of the datasets, respectively.

Considering now the daily mean values, Figure 5 shows the comparison between measured and predicted DNI for the entire period.

As observed in the case of hourly values, the differences between observed and predicted DNI increases from the first day of the forecast to the last one. Another common feature observed is the DNI overestimation for lower values of direct normal irradiance as it can be seen through the trend lines. The statistical indicators obtained are comparable to the analysis made for the hourly values, showing a low forecast bias, with MBE values below 7 W/m$^2$ for all forecast days. Regarding RMSE, an increase of 35% percent (from ~61W/m$^2$ to 76 W/m$^2$) between the first and the last day of forecast. It is evident from the scatter plots of Figure 5 that between roughly 250 and 350 W/m$^2$ the distribution of data points is closer to the y = x line (ratio 1:1), which reveals a good agreement between observations and predictions. The overestimation occurs for observational DNI values below 200 W/m$^2$, with a larger dispersion, which may reflect inaccuracies in IFS cloud representation.

Figure 6 shows the monthly mean of daily values of simulated and measured DNI for the period between 1 August 2018 and 31 July 2019, thus allowing us to analyze the similarity between datasets throughout the year for the different forecast days.

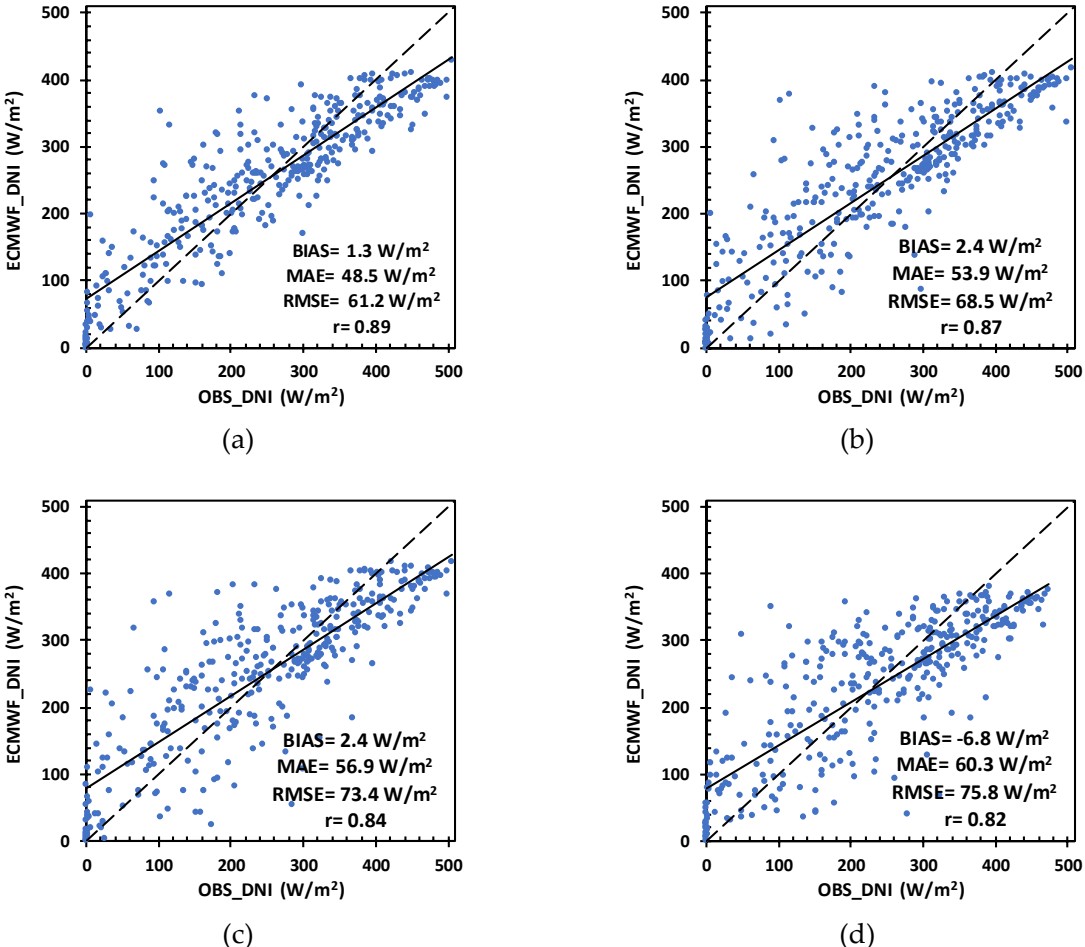

**Figure 5.** Comparison between predicted and measured daily mean DNI for the four prediction days: (**a**) day_0, (**b**) day_1; (**c**) day_2; (**d**) day_3. MBE, MAE, RMSE, and *r* are also presented in each plot. The solid lines are the linear fits and the dashed line represents the y = x line.

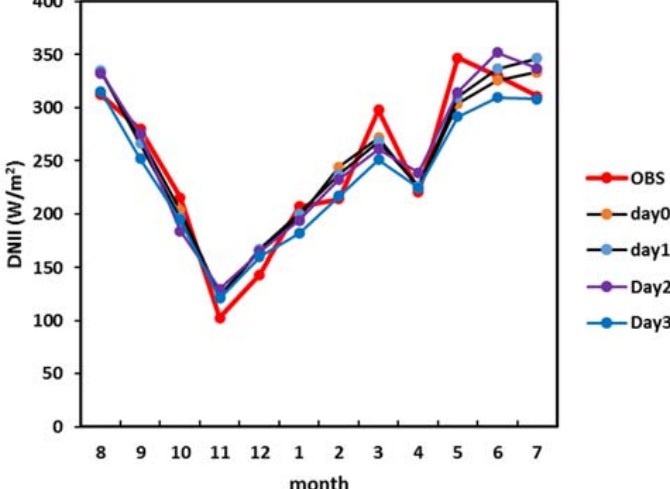

**Figure 6.** Monthly mean of predicted and observed daily mean DNI in Évora for the four different forecast days in the period from August 2018 to July 2019.

As shown in Figure 6, the IFS/ECMWF model overestimates the radiation in more ~50% of the days throughout the year, independently of forecast day, although with small differences datasets.

The variation of statistical indicators between datasets (daily) grouped by months is presented in Figure 7.

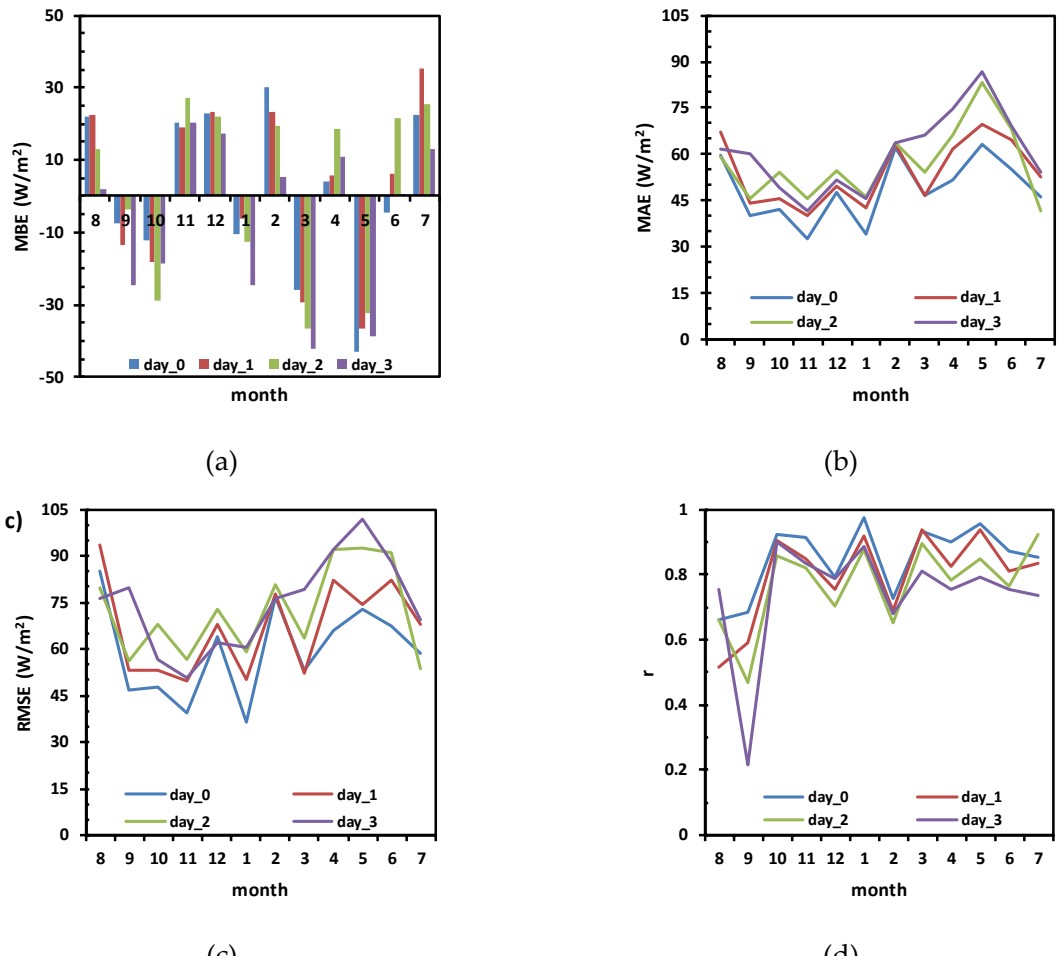

**Figure 7.** Statistical indicators obtained from the comparison between measurements and predictions of daily mean DNI values. (**a**) MBE; (**b**) MAE; (**c**) RMSE; (**d**) correlation coefficient.

Overall, MBE, MAE, and RMSE present better results for the first day of the forecast (day_0). The highest values of statistical errors correspond to the forecasts obtained for day_3.

From Figure 7a, MBE values range from about −42 W/m² to 35 W/m², and show ~60% of the months with positive MBE values (independently of forecast days). According to the same figure, the underestimation occurs in two-thirds of the months belonging to the MMA and SON seasons, probably as a consequence of a less accurate representation of the clouds (or aerosols) at short time scales in the radiative scheme of IFS/ECMWF.

According to Lopes et al. [17], the IFS global model from ECMWF tends to underestimate DNI in clear sky conditions due to the use of a monthly mean profile of aerosols. Perdigão et al. [6] also used the same argument in the assessment and characterization of the shortwave downward radiation incident at the Earth's surface over the Iberian Peninsula using the mesoscale Weather Research and Forecasting (WRF) model.

Figure 7b,c shows that MAE and RMSE present lower values, independently of forecast day, in JJA and SON seasons. MAE and RMSE present a similar variation that as found for MBE (Figure 7b,c) with values ranging from 33 W/m² to 87 W/m² and 36 W/m² to 102 W/m², respectively. As mentioned

above, results show high correlation coefficients for all forecast days, although with the IFS/ECMWF model performing better on the first day of forecast.

The majority of statistical errors found in this work are in line with values obtained in the forecast of DNI by Nonnenmacher et al. [43] and Lara-Fanego et al. [18] using WRF model, and Gala et al. [44] using a clear sky model, among other studies. Table 2 of the article of Law et al. [13], show a summary of the state-of-the-art of DNI accuracy obtained from NWP and other methodologies.

### 3.2. Relation between the DNI Attenuation Index (DAI) and DNI Forecasts

Inaccurate representation of clouds in the radiative transfer scheme of global numerical weather prediction models is the primary cause of errors in the prediction of solar radiation. In this section, the DNI attenuation index (DAI) is proposed to assess and analyze the impact of cloud representation in the DNI forecast from the IFS/ECMWF model, as defined in Section 2.4 (Equation (6)).

Before analyzing the relationship between the DAI index, computed using data from the Évora radiometric station (Section 2.1) and the quality of the DNI forecast errors, the reliability of DAI is assessed, monthly, using the Total Cloud Area Fraction from CERES (Section 2.4) for the same local. According to Almorox et al. [45], global solar irradiation obtained from CERES, monthly, provides very good accuracy for solar radiation studies since their results show a good fit between CERES data and solar radiation data from different meteorological stations over Spain.

The linear regression between CERES cloud fraction and DAI shows a good agreement between these two indexes (Figure 8a), with a correlation coefficient of r ~0.92. When comparing the temporal evolution of DAI and cloud area fraction (CERES) monthly, both time series exhibits a similar pattern (Figure 8b), and shows, as expected, a decrease of cloud cover in the JJA season in contrast with an increase of the cloud cover in the DJF season.

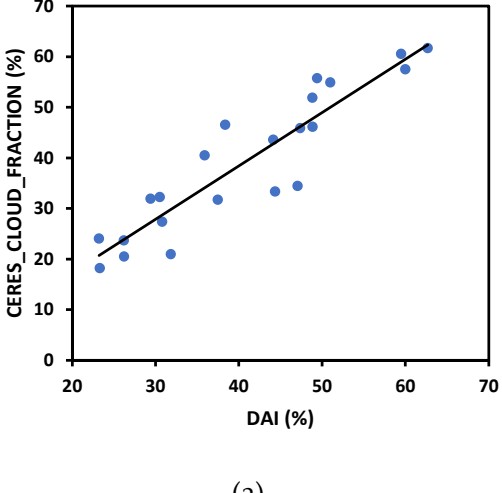

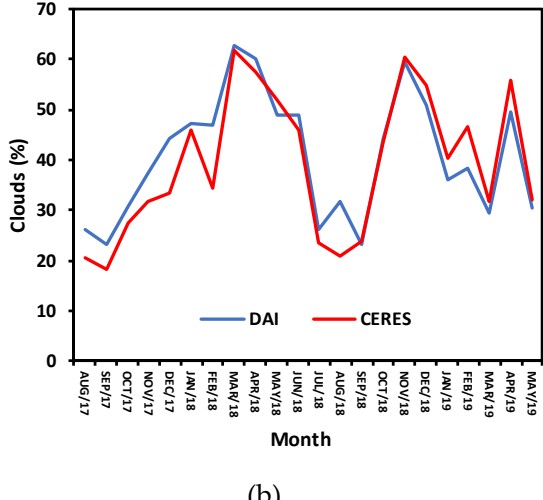

(a)          (b)

**Figure 8.** (**a**) Monthly mean cloud cover from Clouds and the Earth's Radiant Energy System (CERES) versus DNI Attenuation Index (DAI) in Evora, and (**b**) temporal evolution of CERES cloud fraction (red line) and DAI (blue line) in the period between August 2017 and July 2019 (twenty-two months). The black solid line represents the linear fit.

The major discrepancies between datasets occur in February 2018, February 2019, and in August 2018, corresponding to periods in which the region was affected by aerosol events (Saharan dust particles, in February, and forest fires in August).

The results suggest that DAI may be used as a proxy to cloud cover, particularly suitable to estimate the impact of clouds on the DNI forecast.

As for the observations, the DAI of model predictions was also calculated and hereafter is referred to as DAI (IFS). Figure 9 shows a boxplot comparison between DAI and DAI (IFS) index.

From Figure 9 it can be seen that in the majority of analyzed months, the DAI (IFS) is lower than DAI, meaning that the cloud scheme in IFS/ECMWF model underestimates the clouds and aerosols events when compared with the DAI index. DAIs are characterized by a lower variability in summer with more than 50% of the days with values lower than 31.25% (clear sky days) and a higher variability during the spring season (higher IQR values with more than 50% of days with values higher than 31.25%). These results are in line with the study by Royé et al. [46], in which low levels of cloudiness (clear skies days) over the Iberian Peninsula were found for the case of summer months, using satellite data from the MODIS and for the period 2001–2017, except for the Cantabrian coast. On the other hand, the variability of observed DAI is higher, mainly due to a higher variability on actual cloud cover and aerosol concentrations values. This variability also explains the relative high errors reported in the previous section. Perdigão et al. [6] also found high variability in downward shortwave radiation during March.

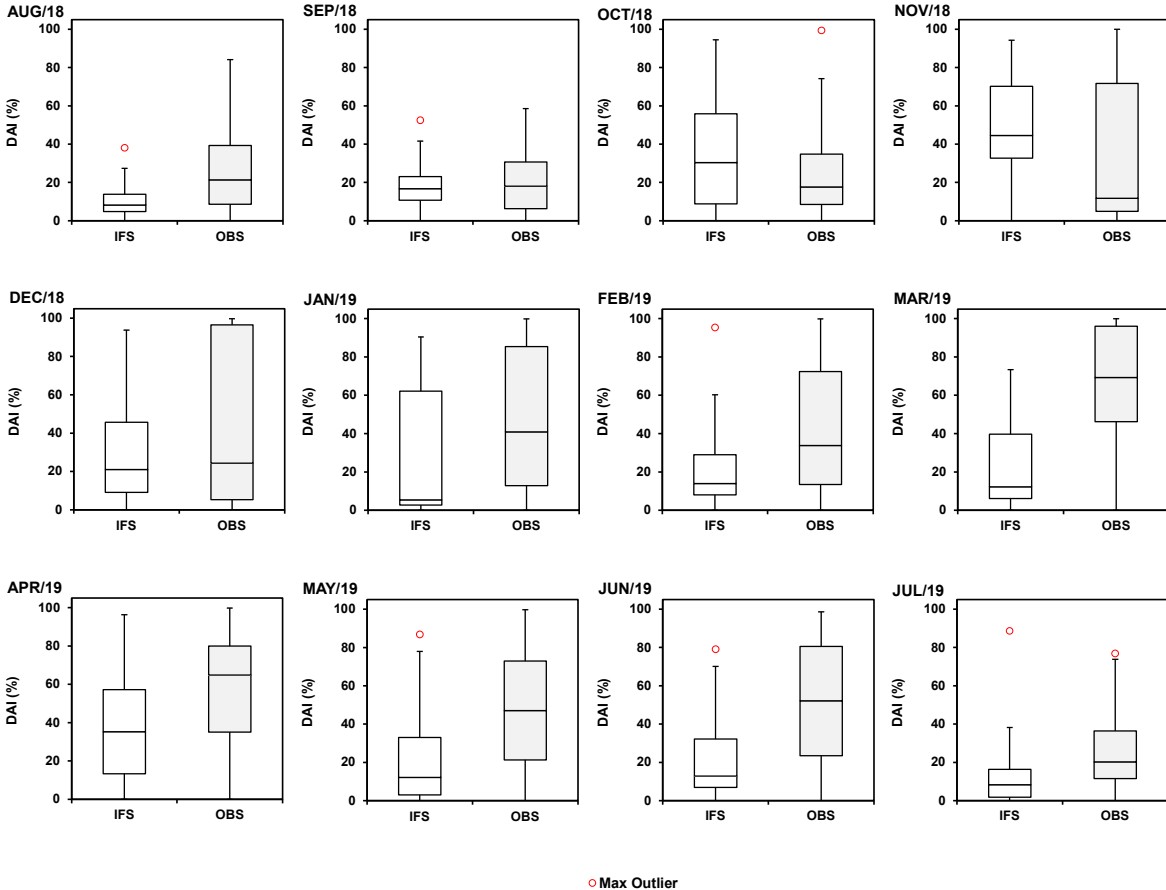

**Figure 9.** Monthly boxplots of daily mean values of DAI based on observations (OBS) and IFS/ECMWF forecasts (IFS), for Évora. The red circles represent outliers (maximum value).

Figure 10 shows the relation between the cloud class, grouped according to cloud coverage (in oktas), estimated based on the DAI, as indicated in Section 2.4, and the statistical indicator RMSE (in three ranges of values), obtained for each day, from hourly raw values. Day type class (I-clear, II-partially cloudy, and III-overcast) is obtained following the WMO guidelines [23].

From both plots of Figure 10, it becomes evident that RMSE strongly depends on the sky conditions, and it is possible to verify that:

(i)     Approximately ~19% of the days present RMSE values lower than 100 $W/m^2$ (blue dots in Figure 10a). This percentage corresponds mostly to a cloud coverage lower than or equal to two oktas-clear skies days;

(ii)   ~47% of the days present a cloud coverage of class II type, in the range (100–200 W/m$^2$);

(iii)   RMSE values above 200 W/m$^2$ occur for ~34% of the days (red dots in Figure 10a). For this value, the majority of days are found in the cloud coverage type II category, suggesting that the model gives worst results in partially cloudy days, due to an inaccurate cloud representation or of their effects on the solar irradiance at the surface. For instance, Lopes [47] found that thin clouds (like cirrus) may cause a decrease in DNI of around 20%;

(iv)   The errors found in summer months can be explained by the monthly constant aerosol climatology used in IFS/ECMWF as argued by Lopes et al. [17].

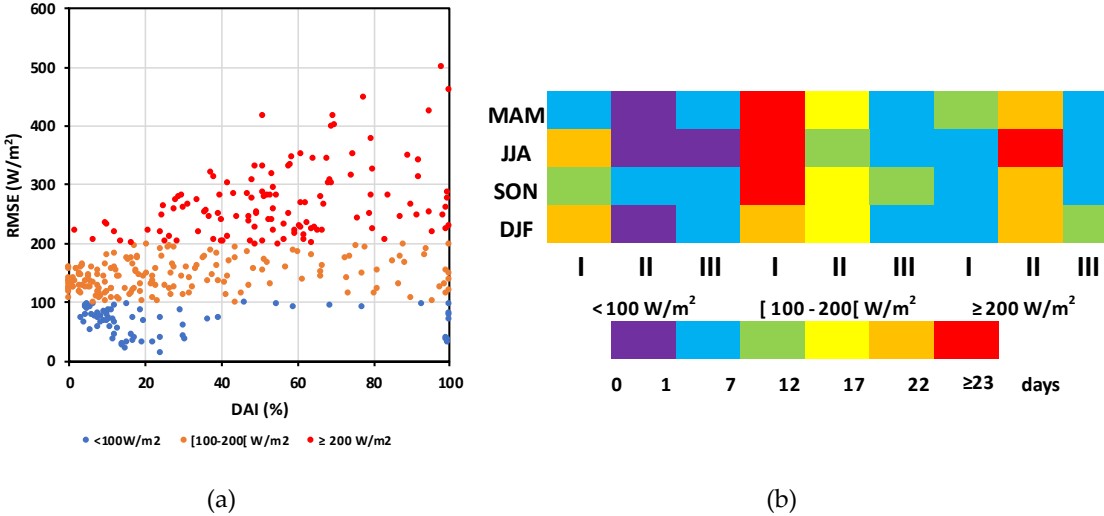

(a)                                         (b)

**Figure 10.** (**a**) Scatter plot of DAI versus Root Mean Square Error (RMSE) for day_0 and (**b**) the number of days in a seasonal basis within different ranges of forecast errors grouped in classes, according to the cloud coverage–class I (0–2 oktas), class II (3–5 oktas), or class III (6–8 oktas) for RMSE.

The analysis of the climatology of cloud cover at Évora based on DAI for the period from 1 August/2018 to 31 July 2019 (Figure 10b), shows that about 45% of the days are in class I, considering only RMSE errors below 200 W/m$^2$, and these days are mainly in the MMA and JJA seasons, when more clear skies days occur over Évora city. These values are consistent with those found by Sanchez-Lorenzo et al. [48] and by Perdigão et al. [6] for the same sky conditions over the Iberian Peninsula.

Concerning the cloud coverage of class II and III, for the period in analysis and independently of the RMSE values, there are ~37% and ~13% of days, respectively. Cloud coverage of type III is mostly found in the DJF season.

It is important to mention that the quality and reliability of the forecast of solar radiation are directly related to the accuracy of cloud representation as well as aerosols. For instance, Lara-Fanego et al. [18] found that RMSE values ranged from 20% to 100% for clear and cloudy skies, respectively, using the WRF model over Andalusia.

Another feature can be seen in Figure 11 where it is possible to observe a relation between the RSR and the DAI for the first day of forecast (day_0).

As expected, from Figure 11, as the DAI increases the RSR increase. The forecast obtained from IFS/ECMWF presents a better performance for clear sky days. In general, the DNI forecast tends to be more accurate (RSR <0.5) for lower values of DAI, which represents ~50% of the days in Évora. This value of clear sky days is of the same order of the study of Freile-Aranda et al. [49], who found, for a climatic region which includes Évora, a minimum cloud cover (at an annual average) around 43%. The best performance (of IFS/ECMWF) are found for summer months (smaller values of RSR ≤0.5 and DAI ≤2 oktas). For instance, Kraas et al. [12], also found that during the summer season the forecasts are generally more reliable than in other seasons.

### 3.3. Statistical Bias Correction Analysis of Daily DNI Forecasts

The bias between solar radiation forecasts from Numerical Weather Prediction models and observations can be decreased by applying a bias post-processing correction. This Bias Correction (BC) methodology have been used in solar radiation studies by several authors, such as Ruiz-Arias et al. [7], Polo et al. [40,49], Perdigão et al. [6], Mejia et al. [41], among other authors.

The forecast values of DNI are corrected following the methodology described in Section 2.5. In Figure 12, the show the result of the application of the MOS method for the various forecast horizons.

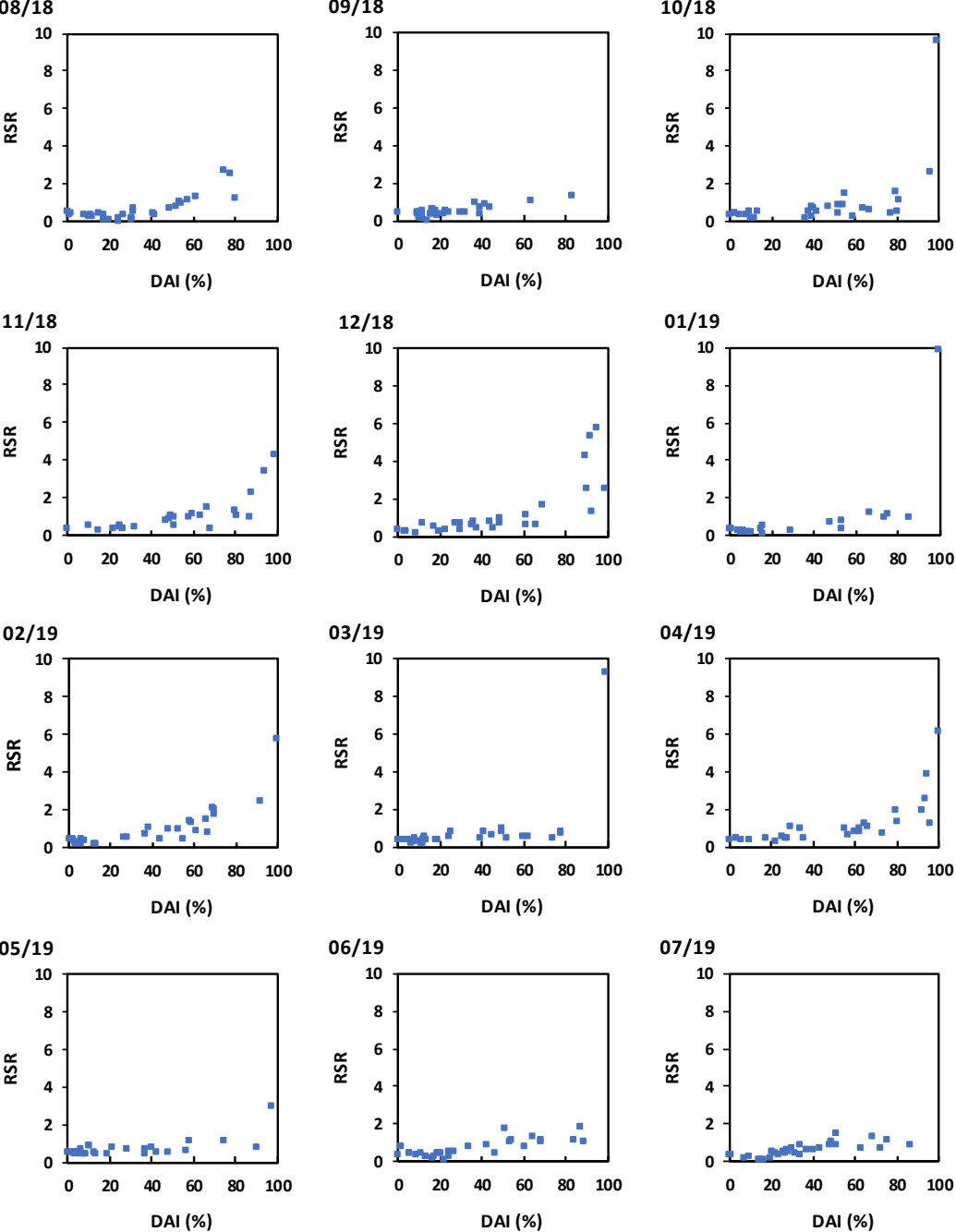

**Figure 11.** Relation between DAI and RMSE-observations standard deviation ratio (RSR) RSR based on hourly mean DNI forecasts and measurements for the first day of predictions between 1 August 2018 and 31 July 2019 (one year). RSR is dimensionless varying between zero and a large positive number.

Error metrics are also presented in graphs using the new corrected predictions.

Overall, MOS correction significantly improves the results. The corrected data exhibited (now) less dispersion around y = x line. Statistical errors decrease in the order of about 30% when compared with initial IFS predictions and independently of the forecast day. For example, MAE values decrease from the interval (49–60 W/m$^2$) to (32–41 W/m$^2$), while RMSE values decrease from (61–76 W/m$^2$) to (43–57 W/m$^2$). The correlation coefficient is in line with previous values, i.e., all *r* values were improved with values ≥0.89.

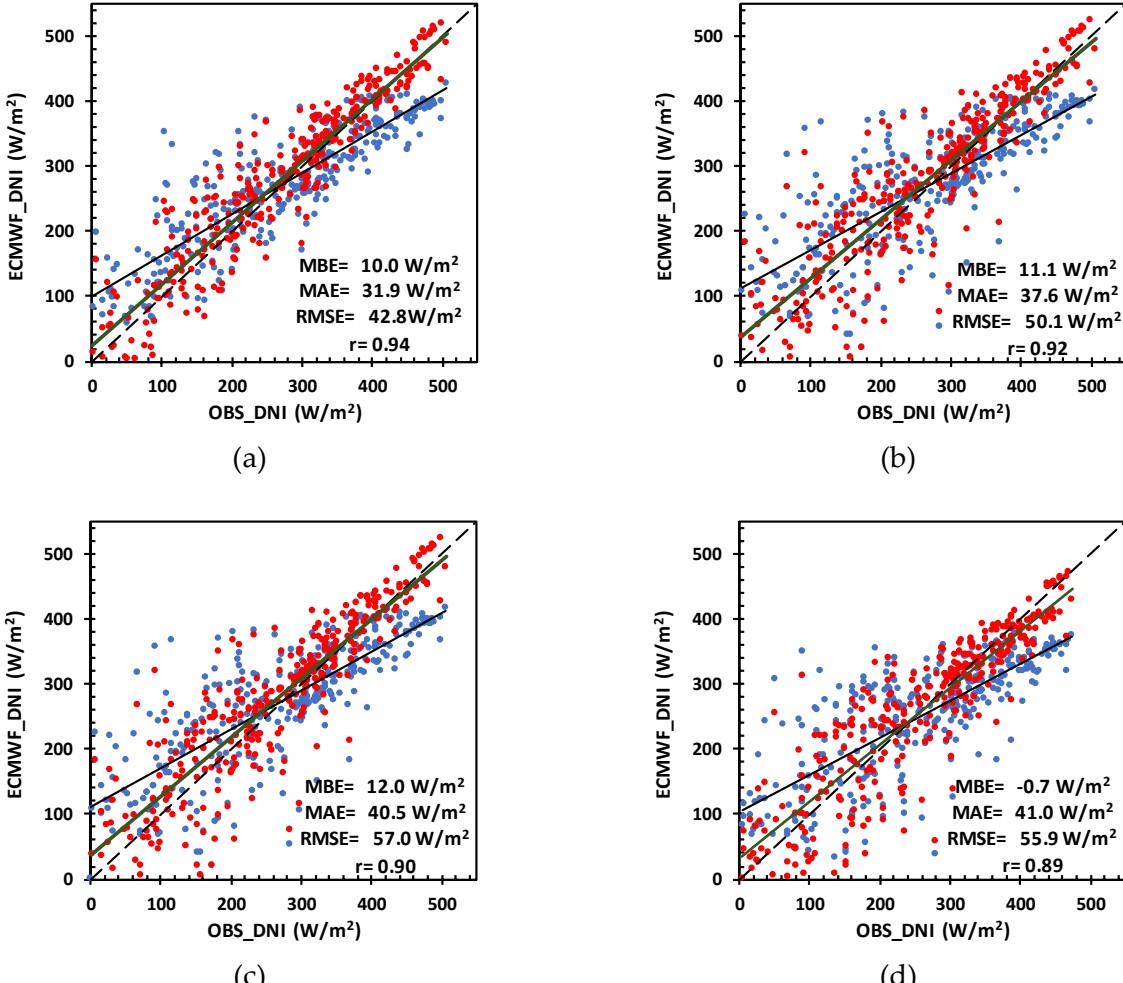

**Figure 12.** Comparison between predicted and measured daily mean DNI for the four prediction days: (**a**) day_0, (**b**) day_1; (**c**) day_2; (**d**) day_3 before (blue dots) and after Bias correction (red dots). MBE, MAE, RMSE, and r, after BC, are also presented in each plot. The solid lines are the linear fits-green for BC procedure and black for IFS/ECMWF raw data-and the dashed line represents the y = x line.

To a better comparison, Figure 13 shows the cumulative distribution functions (CDF) of daily DNI, for each day of forecast, before and after the correction procedure.

Results in Figure 13 show the similarity of the CDF between observational and IFS/ECMWF outputs before and after the bias correction. In general, and independently of the forecast day, as said before, the linear regression method successfully improved the DNI outputs with the new corrected cumulative distribution function plots closer to the observed DNI.

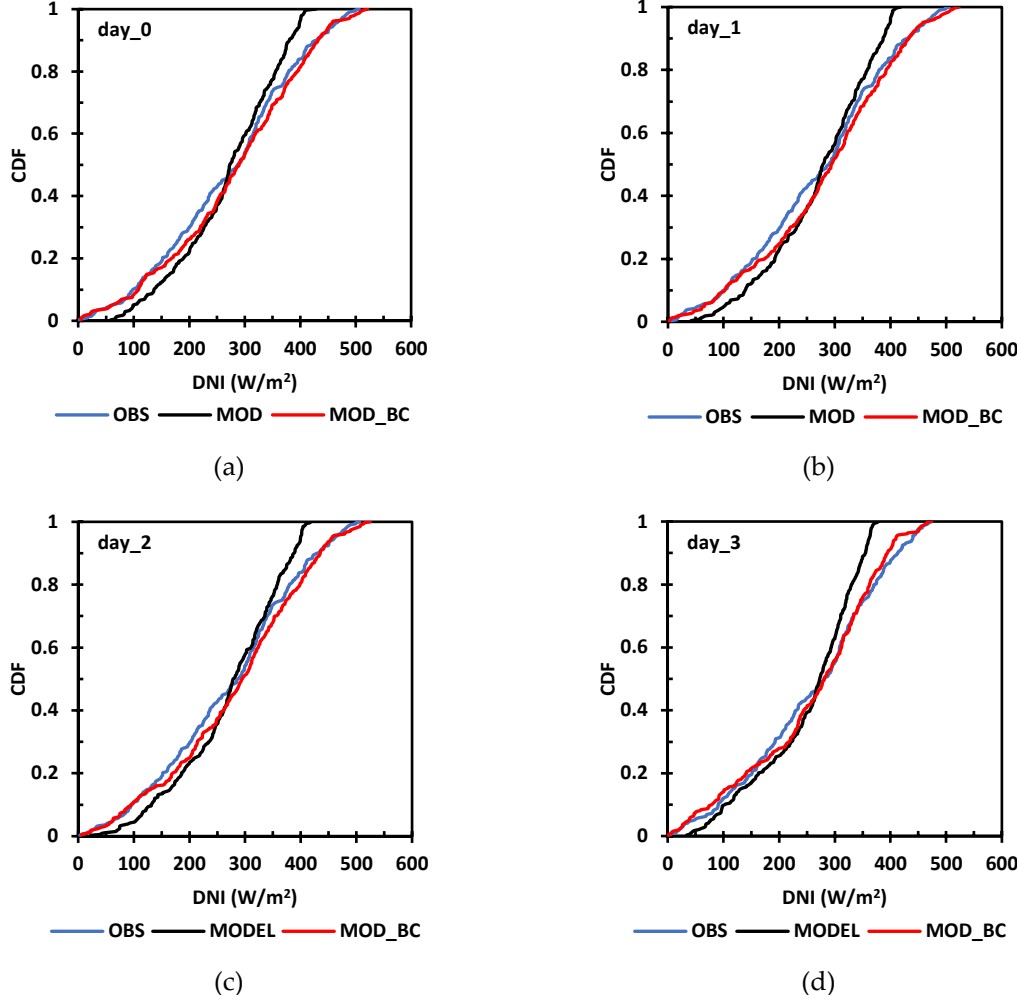

**Figure 13.** Cumulative distribution functions (CDF) of daily mean DNI grouped by day of forecast, (**a**) Day_0; (**b**) day_1; (**c**) day_2; (**d**) day_3, from 1 August 2018 to 31 July 2019, original forecasts (black line), forecasts after Bias Correction (red line) and observations (blue line).

## 4. Conclusions

Given the importance that alternative energies have in a sustainable economic, social, and environmental perspective, it is important to know in advance how solar, wind, or other renewable energy resources change on an hourly, daily, or monthly basis. In this work, DNI from Integrated Forecasting System of European Centre for Medium-Range Weather Forecasts (IFS/ECMWF) dataset was evaluated over one year (1 August 2018 to 31 July 2019) against observed DNI, at hourly time scales, for one station located at Evora (south of Portugal) for different days ahead of forecast (until three days ahead).

Statistical BIAS, MAE, RMSE, RSR, and correlation coefficient were used to assess the relation between IFS/ECMWF and observational DNI. Additionally, this paper, also describes a new methodology based on DNI observations (the called of DNI attenuation index–DAI) to estimate the transparency of the atmosphere in a particular region. The DAI was evaluated with total cloud cover parameter obtained from CERES product data and results showed high correlation coefficients between datasets, suggesting that DAI can be used as proxy to classify the cloud coverage in direct solar radiation studies. This index was used to analyze the relation between the cloud coverage and the predicted DNI, as well as the respective associated error.

The IFS/ECMWF DNI forecasts present similar magnitudes and pattern relatively to observational data, but the errors increase with the forecast lead time (from 1 to 3 days ahead). Discrepancies

between modelled and measured radiation are relatively small mainly for the first three days of forecasts. The analysis of hourly data showed that the DNI is overestimated by IFS/ECMWF. Regarding the correlation coefficient, values are found above ≥0.7, independently of the forecast day. This work also shows that the first day ahead forecast (day_1) has similar error magnitudes in relation to the first 24 h forecast (day_0).

Hourly analysis also shows values of MBE, lower than 20 W/m$^2$. Regarding the MAE and RMSE values, an increase from day_0 to day_3 was observed, with a difference between the first and the third day of forecast ~45% and ~37%, respectively. High correlation coefficients (r ≥0.7) are found for all forecast days.

Daily analysis shows better results, with MBE values lower than 7 W/m$^2$ for all forecast days. In the case of RMSE, values increase about 35% percent from the first day of forecast to the last one. The correlation coefficients of daily data are higher than in the case of hourly data, ranging between 0.82 (day_3) and 0.89 (day_0).

The mean-monthly cloud coverage is well captured by DAI along the year. As expected, the observed DNI is higher in spring and summer months with the lowest values in DAI for the same seasons. The underestimation of cloud cover by the IFS/ECMWF seems to be evident since comparison between observed and predicted DAI reveals that model tends to underestimate the effects of clouds on DNI. This relation was also found for Andalusia (located in Iberian Peninsula) using WRF model by Lara-Fanego et al. [18] in the case of three days ahead DNI forecasts.

The accuracy of IFS/ECMWF to forecast DNI is higher for clear or partially cloudy sky days. DAI index confirms that the performance of the IFS/model decrease with an increasing of clouds/aerosols effects.

A bias correction post-processing through a linear regression was used to correct the IFS/ECMWF predictions, which has shown to significantly improve the forecast for Évora with a decrease in the order of 30% for all statistical error metrics, except for the correlation coefficient, independently of the days ahead in consideration.

The results obtained in this work are consistent with those obtained by Lopes et al. [17] for the same location, and by Nonnenmacher et al. [14], Troccoli, and Morcrette [19], among others, where it was found that errors increase with the lead time forecast. Overall, ECMWF DNI forecasts provide valuable information for the management and operation of CSP plants, especially after the usage of the post-processing bias correction.

**Author Contributions:** Conceptualization, J.P., P.C., R.S. and M.J.C.; formal analysis, J.P.; methodology, J.P., P.C., R.S. and M.J.C.; resources, P.C., R.S. and M.J.C.; supervision, P.C., R.S. and M.J.C.; writing—original draft, J.P.; writing—review and editing, J.P., P.C., R.S. and M.J.C. All authors have read and agreed to the published version of the manuscript.

**Funding:** The work is co-funded by the European Union through the European Regional Development Fund, included in the COMPETE 2020 (Operational Program Competitiveness and Internationalization) through the ICT project (UIDB/04683/2020) with the reference POCI-01-0145-FEDER-007690 and also through the DNI-A (ALT20-03-0145-FEDER-000011) and LEADING (PTDC/CTA-MET/28914/2017) projects.

**Acknowledgments:** The authors acknowledge the ECMWF for providing DNI forecasts through the website and at NCEP Reanalysis data provided by the NOAA/OAR/ESRL PSD, Boulder, Colorado, USA, from their Web site at https://www.esrl.noaa.gov/psd/. The authors also thank the CERES Science Team for the data provided.

**Conflicts of Interest:** The authors declare there is no conflict of interest with regard to this manuscript.

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
