# Peer review of "Assessment of Direct Normal Irradiance Forecasts Based on IFS/ECMWF Data and Observations in the South of Portugal"

_forecasting, doi:10.3390/forecast2020007_

Round 1

Reviewer 1 Report

This is a good paper that should be published once some aspects are clarified.

First, it is stated on page 4 that the RMSE represents non-systematic errors. However, as defined in equation (3), RMSE is measuring both systematic bias error and non-systematic errors. So either state this or change equation (3) so that the systematic error is subtracted.

Second, there are several problems with equation (10). The subscript j is absent from m and x and, if equation (9) is correct, shouldn't  the bias corrected model result  be (ymo/m -b)? 

Third, on page 12 figure 8, the graphs  referred to as (a) and (b) should be switched and on page 13 figure (9), it is impossible for me to understand what the red o refers to because the figure caption is not clear. Why does the dot appear in some panels and not others. Why does the dot appear above IFS in some panels, OBS or both in other panels?

These ambiguities should be cleaned up before publication of the paper.

Author Response

-This is a good paper that should be published once some aspects are clarified.

The authors greatly acknowledge the anonymous reviewers for carefully reading the manuscript and providing constructive comments. The minor comments raised by the reviewer are addressed below.

-First, it is stated on page 4 that the RMSE represents nonsystematic errors. However, as defined in equation (3), RMSE is measuring both systematic bias error and non-systematic errors. So either state this or change equation (3) so that the systematic error is subtracted.

We have rephrased these sentences, which now read as: “The MBE represents a systematic error between predicted and observational values, while the RMSE quantifies the spread in the distribution of errors.” (line 168)

-Second, there are several problems with equation (10). The subscript j is absent from m and x

The reviewer is right. We insert the subscript j in equation 10.

 -if equation (9) is correct, shouldn't the bias corrected model result be (y /m -b)?

The equation is correct. In this work we used the same methodology as explained in article of Polo et al. (2016). [https://doi.org/10.1016/j.solener.2016.03.001]. We add this reference.

 -Third, on page 12 figure 8, the graphs referred to as (a) and (b) should be switched and on page 13 figure (9).

Done. We have now corrected the figure caption. (line 361)

-It is impossible for me to understand what the red o refers to because the figure caption is not clear. Why does the dot appear in some panels and not others. Why does the dot appear above IFS in some panels, OBS or both in other panels?

The legend was missing. We have now corrected the figure (with legend) and added in figure caption: “The red circles represent outliers (maximum value).”

Reviewer 2 Report

This paper deals with the assessment of the irradiance forecasts provided by ECWMF at the city of Evora, Portugal. The topic is interesting since accurate solar forecasts are of high importance for the integration of this solar power in the electrical grid in safety conditions. Understanding the uncertainty in solar forecasts is a fundamental step for an effective use of this information in solar generation systems. The main weaknesses of the paper are related to the use of only one site to perform the validation and the absence of any novel procedure. However, the paper is well-structured and provide valuable information to be used as reference for other studies on the field. Thus, I recommend the publication with minor changes:

- In the introduction, only the usefulness of solar forecasts for CSP plants is mentioned. The solar forecasts are also important in photovoltaic (PV) plants. Market operations, PV plant control and grid stability requires accurate estimations of solar irradiance from seconds to days.

- Line 90: “solar parks”, replace by “solar plants”

- Figures: It would be suitable the insertion of a legends in figures. In most cases the represented data is explained in the caption.

Author Response

-This paper deals with the assessment of the irradiance forecasts provided by ECWMF at the city of Evora, Portugal. The topic is interesting since accurate solar forecasts are of high importance for the integration of this solar power in the electrical grid in safety conditions. Understanding the uncertainty in solar forecasts is a fundamental step for an effective use of this information in solar generation systems. The main weaknesses of the paper are related to the use of only one site to perform the validation and the absence of any novel procedure. However, the paper is well-structured and provide valuable information to be used as reference for other studies on the field. Thus, I recommend the publication with minor changes:

We really appreciate the reviewer’s comments and his/her effort to review our manuscript. The one site used in this study was chosen for two reasons: there are several projects for the construction of CSP plants in the Évora region; We have access to the data of this station, whose reliability and quality has been recognized in many works (e.g. Cavaco et al., 2018, Pereira et al., 2019; Lopes et al., 2018).

 The minor comments raised by the reviewer are addressed below.

 -In the introduction, only the usefulness of solar forecasts for CSP plants is mentioned. The solar forecasts are also important in photovoltaic (PV) plants. Market operations, PV plant control and grid stability requires accurate estimations of solar irradiance from seconds to days.

Accurate forecast of solar components at a given location is of great importance for projects related with solar energy, either PV and CSP. However, the DNI forecast, the main objective of this work, is more useful for CSP power plants. Nevertheless, following the suggestion of the reviewer, we added the following paragraph and two new references in the manuscript (line 47):

 Concerning solar energy, there are two main ways of converting solar energy into electricity: photovoltaic (PV) and concentrating solar power (CSP). The PV panels convert either direct and diffuse solar irradiance, while the CSP technology only concentrate the Direct Normal Irradiance (DNI). The focus of this work is on the prediction of DNI, in view of its use in CSP plant management. The forecast of global solar radiation (direct + diffuse) for the same region was addressed, for example, in [6] and [9].

 Reference [9]: Pereira, S., Canhoto, P., Salgado, R., Costa, M.J.,. Development of an ANN based corrective algorithm of the operational ECMWF global horizontal irradiation forecasts. Solar Energy 2019, 185, 387–405, https://doi.org/10.1016/j.solener.2019.04.070

- Line 90: “solar parks”, replace by “solar plants”

Done. (line 96)

 - Figures: It would be suitable the insertion of a legends in figures. In most cases the represented data is explained in the caption.

Figures2 and 9 have been altered according to the reviewer's suggestion.
